# Prevalence and determinants of depression, anxiety, and stress among secondary school students

**Tingting Dong** [1☯*], **Yumei Wang** [2☯], **Yuanjie Lin** [3]

**1** Department of Nursing, Hospital of Chengdu Office of People's Government of Tibetan Autonomous Region, Chengdu, China, **2** College of Materials Science and Engineering, Xihua University, Chengdu, China, **3** School of Public Health, Sichuan University, Chengdu, China

☯ These authors contributed equally to this work.
* 542432958@qq.com

## Abstract

### Background

By investigating the depression, anxiety and stress status of secondary school students, we further analysed the influencing factors of the changes in depression, anxiety and stress among secondary school students, so as to provide references for subsequent studies and interventions on adolescent mental health.

### Methods

This study was a cross-sectional research study conducted in a city in Southwest China. A questionnaire survey was conducted during March-June 2022 among students attending secondary schools within a city. The surveyed secondary schools were not categorized as public or private. It mainly investigated the general information of middle school students, Depression-Anxiety-Stress Scale (DASS-21), and assessed the mental health status of middle school students.

### Results

A total of 2,716 secondary school students were surveyed, 1,230 males (45.29%) and 1,486 females (54.71%). Among them, 24.4% of the secondary school students had depressive symptoms, 41.4% had anxiety symptoms, and 15.6% had stress; Through logistic regression analysis, attending high school, ranking <60% in academic performance, little or no family psychological support and peer psychological support were the main influencing factors.

**Data availability statement:** All survey data files are now accessible via the OSF database. The access link and registered DOI are: https://doi.org/10.17605/OSF.IO/GQUBA.

**Funding:** This study was supported by the Chengdu Medical Association Research Project (Project No. 2024350).

**Competing interests:** The authors have declared that no competing interests exist.

## Conclusion

The present study demonstrated that the high school student population, students with an academic ranking of <60%, and students who lacked family/peer psychological support were at higher risk for mental health. These findings provide an empirical basis for developing graded intervention strategies for specific risk factors..

## Introduction

Adolescence, defined by the World Health Organisation (WHO) as the age of 10–19 years [1], is a critical window for the development of mental health. In the Chinese education system, junior high school includes the junior (grades 7–9, usually 12–15 years old) and senior (grades 10–12, 16–18 years old) stages, and the Junior High School Academic Level Examination (JHSE) and the National Unified Examination for Admission to General Tertiary Institutions (Gao Kao) bring different academic pressures to students [2]. Recent evidence suggests that key risk factors, including academic stress, family relationship dynamics, and peer competition, are entrenched in early adolescence [3], and longitudinal studies have shown that 68% of adult psychiatric disorders originate before the age of 18 [4].In the Global Burden of Disease Survey, depressive disorders accounted for the largest share, at 37.3%, followed by anxiety disorders at 22.9% [5], adding to the burden of disease on society and families.

Depression, generalized anxiety and stress are the most common mental health problems among adolescents [6]. Mojtabai et al. [7] reported that the prevalence of major depressive episodes among U.S. adolescents increased from 8.7% to 11.3% from 2005 to 2014, and that middle school students are at a psychologically unstable stage of their lives, with a high prevalence of mental illness [8]. A research study in which 1,381 Indonesian secondary school students participated showed that 60% of secondary school students showed symptoms of depression [9]. The prevalence of depression among high school students in China is about 27.12% and shows a rising trend. The psychological problems of secondary school students in China are becoming more and more prominent [10–12], which negatively affects the healthy growth of secondary school students; therefore, a wider understanding of the causes of depression, anxiety and stress in secondary school students should be developed.

According to data from the Global Burden of Disease Survey study, the majority of mental health disorders among adolescents go unrecognized (and therefore untreated), accounting for 13% of the global burden of disease [5]. Mental health problems in adolescents can lead to substance abuse, self-harm, and suicidal behavior [13,14], while having a serious negative impact on their physical health, academic performance, and socialization [15,16]. Additionally, mental health disorders not only occur at a younger developmental age, but also continue to exist and affect them in later life [17]. Therefore, this study aims to understand the depression, anxiety and stress status of secondary school students and to analyse the factors influencing

changes in depression, anxiety and stress in secondary school students, so as to provide a reference basis for subsequent research and intervention in adolescent mental health.

## Method

### Setting

This is a cross-sectional research study conducted in Ya'an City, Southwest China. A questionnaire survey was conducted during March-June 2022 among students attending middle schools within Ya'an City. The surveyed secondary schools were not categorized as public or private. The study was approved by the Ethics Committee of the Hospital of the People's Government of the Tibet Autonomous Region in Chengdu (Ethics Approval No. 50, 2022). Written informed consent was obtained from the students and their guardians for this study.

### Inclusion and exclusion criteria

This study was conducted from March to June 2022 in four middle schools in Ya'an City, Sichuan Province, using convenience sampling method. According to the Chinese standard for dividing education stages, junior high school corresponds to grades 7–9 (age 12–15 years old) and senior high school corresponds to grades 10–12 (age 16–18 years old). Inclusion criteria (1) students enrolled in school; (2) obtaining dual informed consent from themselves and their guardians; and (3) being able to complete the Chinese electronic questionnaire independently. Exclusion criteria: those who refused to participate in the study for various reasons.

### Sampling

The sample size for a cross-sectional study is calculated using the following formula: $N = \frac{Z^2_{1-\frac{\alpha}{2}} \, p(1-p)}{d^2}$. In this study, a cluster sampling method is employed. As the expected prevalence of mental disorders among adolescents is unknown, we assume P = 0.5, d = 0.05, α = 0.05, yielding Z1-α/2 = 1.96. Based on this, the calculated sample size is 384. Considering the design effect (DE) of cluster sampling (DE = 4), the adjusted sample size is 1,536. To account for potential attrition during follow-up, we further increase the sample size by 20%, resulting in a final required sample size of 1,844.

### Survey questionnaire

**General information questionnaire.** This study was designed by all the researchers according to the purpose of the study, we used questionnaires to collect gender, grade, school accommodation, area of residence, number of co-residents, initiative to talk about the problems encountered, the person to whom the problem was confided and family residence population to ensure the accuracy of the data; academic results were obtained directly from the uniform standardized test papers at the end of the previous semester in the school, aiming to protect the privacy of the students and at the same time, to ensure the scientificity and validity of the study.

**Depression, anxiety and stress scale (DASS-21).** The short-form Depression Anxiety Stress Scales-21 (DASS-21) developed by Loivdband [18] was used to assess negative emotions in secondary school students. The scale consists of 21 items, each of which is rated on a 4-point scale ranging from 0 (does not apply to me at all) to 3 (applies to me very much or most of the time). The score for each subscale is calculated by summing the scores of the items that make up the subscale. Higher scores indicate greater severity of the corresponding mood state, and the DASS-21 scale provides cut-off points for different levels of symptom severity in order to categorise an individual's mood state. Depression Scale: 0–9 is normal range, 10–13 is mild depression, 14–20 is moderate depression, and 21 and above is severe or very severe depression. Anxiety Scale: 0–7 is normal range, 8–9 is mild anxiety, 10–14 is moderate anxiety, 15 and above is severe or very severe anxiety. Stress subscale: 0–14 is normal range, 15–18 is mild stress, 19–25 is moderate stress, and 26 and above is severe or very severe stress. the factor structure of DASS-21 is stable, and the scale can be used in the Chinese population [19,20], with good convergent and discriminant validity and high internal consistency.

## Data collection

The questionnaire was developed using an online survey platform called "QuestionStar", and two uniformly trained surveyors were trained and assessed before the survey began. After passing the assessment, the investigator distributed the questionnaire to the school's classroom teachers via email and WeChat (a popular Chinese social application). The classroom teacher will then send the questionnaire's webpage and invitation to students to complete. If participants had any questions about the questionnaire, they could contact the investigators via email, phone or WeChat. Each participant had the right to decide whether or not to participate in the study and could withdraw from the study at any time. Prior to submitting the content of the questionnaire, a page will be displayed showing the objectives of this study and seeking informed consent from the participants. Participants could continue to complete the questionnaire if they provided informed consent. In addition, consent was also obtained from the parents of the participants before the study began the questionnaire, and the parents of the participants could have their children refuse to participate at any time. Each questionnaire was checked for logical errors and missing items by the principal investigator and two investigators. A total of 2900 questionnaires were distributed in this study and 2716 complete questionnaires were recovered with an overall validity rate of 93.66%.

## Data analysis

For continuous variables, data characteristics were described using the mean ($\bar{x}$) and standard deviation (SD), while categorical variables were presented as frequencies (n) and percentages (%). For continuous variables, if the data followed a normal distribution, a t-test was used to compare differences between groups with varying psychological conditions; otherwise, the Wilcoxon rank-sum test was applied. Categorical variables were compared using the chi-square test. To further explore the sources of differences in multicategory variables, post-hoc tests were performed with Bonferroni correction for p-values. A binary logistic regression model was employed to examine the effects of various factors on mental health. Initially, univariate binary logistic regression analysis was conducted (Model 1). Based on the results of Model 1, to address multicollinearity, variables with a p-value greater than 0.1 or a variance inflation factor (VIF) exceeding 10 were excluded from the analysis (Model 2). The goodness-of-fit of the models was evaluated using the R-squared ($R^2$) statistic. All statistical analyses were performed using R software, and a two-sided p-value $< 0.05$ was considered statistically significant.

## Results

### Analysis of factors influencing DASS-21 in middle school students

As shown in Table 1, high school students had significantly higher rates of depression (76.5% vs. 23.5% for junior high school students), anxiety (73.8% vs. 26.2%), stress (74.7% vs. 25.3%) were significantly higher (p < 0.001 or p = 0.016); residential learning students exhibited higher rates of depression (69.5% vs. 30.5% for commuting learning students, p = 0.014) and anxiety (76% vs. 24%, p = 0.008) and stress risk (29.8% vs. 70.2%); village/rural students had significantly higher anxiety detection rates than city students (60.8% vs. 39.2%, p < 0.001). Notably, students from single-parent families had significantly higher risks of depression, anxiety, and stress (depression, anxiety p < 0.001, stress p = 0.002). Students who reported 'less' family or peer support, those who passively sought help, and those without a confidant had significantly higher proportions of all three symptoms (p < 0.001 for all), while those who actively sought help and those with family or peer confidants had lower symptom risks.

### DASS-21 scores and distribution characteristics

A total of 2,716 secondary school students completed the online survey, 1,230 (45.29%) males and 1,486 (54.71%) females. As shown in Table 2, most of the students had normal levels of depression, anxiety and stress but still 663 (24.4%) had depressive symptoms, 1125 (41.4%) had anxiety symptoms and 423 (15.6%) had stress. In the depression subtest, 1.17% of the students were extremely depressed, 1.17% were severely depressed, 8.47% were moderately

**Table 1. Analysis of factors influencing DASS-21 in middle school students.**

| Variables | Levels | Depression[a] | | | Anxiety[a] | | | Stress[a] | | |
|---|---|---|---|---|---|---|---|---|---|---|
| | | No (N = 2053) | Yes (N = 663) | P Values | No(N = 1591) | Yes (N = 1125) | P Values | No (N = 2293) | Yes (N = 423) | P Values |
| **Demographics** | | | | | | | | | | |
| Age | Mean ± SD | 15.0 ± 1.8 | 15.5 ± 1.7 | <0.001 | 14.9 ± 1.7 | 15.4 ± 1.8 | <0.001 | 15.1 ± 1.8 | 15.4 ± 1.7 | 0.003 |
| Gender | Male | 934 (45.5%) | 296 (44.6%) | 0.736 | 736 (46.3%) | 494 (43.9%) | 0.241 | 1036 (45.2%) | 194 (45.9%) | 0.837 |
| | Female | 1119 (54.5%) | 367 (55.4%) | | 855 (53.7%) | 631 (56.1%) | | 1257 (54.8%) | 229 (54.1%) | |
| Household size(n) | 1 | 34 (1.7%) | 29 (4.4%) | <0.001 | 21 (1.3%) | 42 (3.7%) | <0.001 | 45 (2%) | 18 (4.3%) | 0.002 |
| | 2 | 234 (11.4%) | 81 (12.2%) | | 181 (11.4%) | 134 (11.9%) | | 252 (11%) | 63 (14.9%) | |
| | 3 | 650 (31.7%) | 189 (28.5%) | | 505 (31.7%) | 334 (29.7%) | | 724 (31.6%) | 115 (27.2%) | |
| | ≥4 | 1135 (55.3%) | 364 (54.9%) | | 884 (55.6%) | 615 (54.7%) | | 1272 (55.5%) | 227 (53.7%) | |
| Domicile | Village/Rural | 1117 (54.4%) | 393 (59.3%) | 0.032 | 826 (51.9%) | 684 (60.8%) | <0.001 | 1269 (55.3%) | 241 (57%) | 0.57 |
| | City | 936 (45.6%) | 270 (40.7%) | | 765 (48.1%) | 441 (39.2%) | | 1024 (44.7%) | 182 (43%) | |
| **Academic Factors** | | | | | | | | | | |
| Grade | Junior High | 669 (32.6%) | 156 (23.5%) | <0.001 | 530 (33.3%) | 295 (26.2%) | <0.001 | 718 (31.3%) | 107 (25.3%) | 0.016 |
| | Senior High | 1384 (67.4%) | 507 (76.5%) | | 1061 (66.7%) | 830 (73.8%) | | 1575 (68.7%) | 316 (74.7%) | |
| Learning styles | commuter learning[b] | 524 (25.5%) | 202 (30.5%) | 0.014 | 456 (28.7%) | 270 (24%) | 0.008 | 600 (26.2%) | 126 (29.8%) | 0.137 |
| | residential learning[c] | 1529 (74.5%) | 461 (69.5%) | | 1135 (71.3%) | 855 (76%) | | 1693 (73.8%) | 297 (70.2%) | |
| grade ranking | Top 10% | 307 (15%) | 76 (11.5%) | <0.001 | 222 (14%) | 161 (14.3%) | 0.824 | 323 (14.1%) | 60 (14.2%) | 0.077 |
| | 11%−30% | 637 (31%) | 173 (26.1%) | | 465 (29.2%) | 345 (30.7%) | | 701 (30.6%) | 109 (25.8%) | |
| | 31%−60% | 663 (32.3%) | 215 (32.4%) | | 522 (32.8%) | 356 (31.6%) | | 743 (32.4%) | 135 (31.9%) | |
| | < 60% | 446 (21.7%) | 199 (30%) | | 382 (24%) | 263 (23.4%) | | 526 (22.9%) | 119 (28.1%) | |
| **Psychosocial Factors** | | | | | | | | | | |
| Family Support | seldom | 79 (3.8%) | 75 (11.3%) | <0.001 | 64 (4%) | 90 (8%) | <0.001 | 101 (4.4%) | 53 (12.5%) | <0.001 |
| | average | 756 (36.8%) | 332 (50.1%) | | 562 (35.3%) | 526 (46.8%) | | 893 (38.9%) | 195 (46.1%) | |
| | More | 1218 (59.3%) | 256 (38.6%) | | 965 (60.7%) | 509 (45.2%) | | 1299 (56.7%) | 175 (41.4%) | |
| Peer Support | seldom | 59 (2.9%) | 59 (8.9%) | <0.001 | 48 (3%) | 70 (6.2%) | <0.001 | 76 (3.3%) | 42 (9.9%) | <0.001 |
| | average | 506 (24.6%) | 206 (31.1%) | | 359 (22.6%) | 353 (31.4%) | | 581 (25.3%) | 131 (31%) | |
| | More | 1488 (72.5%) | 398 (60%) | | 1184 (74.4%) | 702 (62.4%) | | 1636 (71.3%) | 250 (59.1%) | |
| Help-seeking | Passive | 97 (4.7%) | 114 (17.2%) | <0.001 | 77 (4.8%) | 134 (11.9%) | <0.001 | 132 (5.8%) | 79 (18.7%) | <0.001 |
| | Reactive | 900 (43.8%) | 373 (56.3%) | | 666 (41.9%) | 607 (54%) | | 1053 (45.9%) | 220 (52%) | |
| | Proactive | 580 (28.3%) | 118 (17.8%) | | 455 (28.6%) | 243 (21.6%) | | 614 (26.8%) | 84 (19.9%) | |
| | Low need | 476 (23.2%) | 58 (8.7%) | | 393 (24.7%) | 141 (12.5%) | | 494 (21.5%) | 40 (9.5%) | |
| Preferred confidant | No confidant | 101 (4.9%) | 145 (21.9%) | <0.001 | 73 (4.6%) | 173 (15.4%) | <0.001 | 148 (6.5%) | 98 (23.2%) | <0.001 |
| | Family | 569 (27.7%) | 95 (14.3%) | | 458 (28.8%) | 206 (18.3%) | | 610 (26.6%) | 54 (12.8%) | |
| | Peers | 669 (32.6%) | 229 (34.5%) | | 501 (31.5%) | 397 (35.3%) | | 744 (32.4%) | 154 (36.4%) | |
| | Specific close person | 187 (9.1%) | 96 (14.5%) | | 144 (9.1%) | 139 (12.4%) | | 217 (9.5%) | 66 (15.6%) | |
| | Low need | 527 (25.7%) | 98 (14.8%) | | 415 (26.1%) | 210 (18.7%) | | 574 (25%) | 51 (12.1%) | |

[a]. DASS-21 cutoff scores: Depression ≥10, Anxiety ≥8, Stress ≥15.

[b]. commuter learning: commuting learning students denote pupils who return home daily, residing within feasible commuting distance from school.

[c]. residential learning: In-school residence students under resource-limited settings refer to those who reside at school dormitories due to geographical constraints or educational policy implementations (e.g., school consolidation programs in rural areas).

**Table 2. DASS-21 scores and distribution characteristics.**

|  | Normal(%) | Mild(%) | Moderate(%) | Severe(%) | Extremely severe (%) | Total (%) |
|---|---|---|---|---|---|---|
| **Depression** | 2053(75.59) | 337(12.41) | 230(8.47) | 48(1.77) | 48(1.77) | 2716(100) |
| **Anxiety** | 1591(58.58) | 397(14.62) | 499(18.37) | 111(4.09) | 118(4.34) | 2716(100) |
| **Stress** | 2293(84.43) | 233(8.58) | 114(4.2) | 58(2.14) | 18(0.66) | 2716(100) |

depressed, and 12.41% were mildly depressed. On the anxiety subtest, 4.34% of students were extremely anxious, 4.09% were severely anxious, 18.37% were moderately anxious, and 14.62% were mildly anxious. In the stress subtest, 0.66% of the students were in the extremely severe stress category, 2.14% were in the severe stress category, 4.2% were in the moderate stress category, and 8.58% were in the mild stress category.

### Logistic regression analysis of DASS-21 in secondary school students

This study used logistic regression to analyse the factors influencing the mental health of 2,716 secondary school students (DASS-21 scale). After controlling for confounding variables (Model 2), the results showed that adequate family support significantly reduces the risk of depression (OR = 0.45, 0.30–0.68, $p < 0.001$); those who actively seek help have only 48% of the depression risk of those who passively seek help (OR = 0.48, 0.32–0.73); students who actively confide in their families have a significantly lower risk of depression (OR = 0.22, 0.15–0.33, $p < 0.001$), anxiety (OR=0.27, 0.19–0.4, $p < 0.001$), and stress (OR=0.23, 0.15–0.36, $p < 0.001$). High school students had significantly higher anxiety levels than junior high school students (OR=0.1, 0.06–0.17, $p < 0.001$). The risk of depression remained significantly elevated among students in the bottom 40% of academic rankings (OR = 1.62, 1.17–2.25, $p = 0.004$). Middle school students from single-parent families had increased risks of depression and anxiety ($p < 0.05$); gender did not show significant effects in any of the models ($p > 0.05$) (Table 3).

## Discussion

This study surveyed a total of 2,716 secondary school students. Based on the results of the DASS-21 questionnaire, the prevalence rates for depressive symptoms, anxiety symptoms, and stress symptoms were 24.41% (n = 663), 41.42% (n = 1,125), and 15.57% (n = 423), respectively. Logistic regression analysis indicated that attending high school, living in a dormitory long-term, having a single-parent family structure, academic performance ranking below the 60th percentile in the class, lack of effective support from family and peers, and low willingness to seek help were common significant predictors of depressive, anxiety, and stress symptoms.

This study reveals that the prevalence rates of depressive symptoms, anxiety symptoms, and stress symptoms among high school students are significantly higher than those among junior high school students, with this disparity rooted in the multi-dimensional pressure mechanisms of the college entrance examination system. According to relevant research data, over 10 million high school graduates take the Gaokao annually, competing for 6–8 million university spots through the centralised university admission system [21]. From 2017 to 2021, China's gross enrolment rate in higher education accounted for 45.7% to 57.8% of the population in that age group [22]. As the core method for selecting talent in higher education, the Gaokao has led to high school students spending an average of 11.2 hours per day on studying (an increase of 31.8% compared to junior high school students) [23], continuously activating the hypothalamic-pituitary-adrenal axis and increasing the risk of anxiety by 32.7% [24]. In key Gaokao provinces like Jilin, where the first-tier university admission rate is 20.3%, 61.7% of students set the goal of attending a 'Double First-Class University,' with 70.2% experiencing clinical anxiety due to the gap between their goals and reality [25]. The cumulative effects of the family-school system are particularly pronounced, 82.4% of parents have explicit academic expectations, and 48.6% expect their children to attend key universities, leading to 60.3% of students experiencing 'fear of letting down others' [26]. Additionally, a daily schedule of 12 hours (7:00 AM to 9:00 PM) and

**Table 3. Logistic regression analysis of DASS-21 in secondary school students.**

| Variables | Levels | Depression Model 1[a] | Depression Model 2[b] | Anxiety Model 1[a] | Anxiety Model 2[b] | Stress Model 1[a] | Stress Model 2[b] |
|---|---|---|---|---|---|---|---|
| **Demographics** | | | | | | | |
| Age | | 1.16 (1.10–1.22, p<0.001) | 1.24 (1.08–1.43, p=0.003) | 1.19 (1.14–1.25, p<0.001) | 2.06 (1.79–2.38, p<0.001) | 1.10 (1.03–1.17, p=0.003) | 1.08 (0.91–1.27, p=0.371) |
| Gender | Male | ref | | ref | | ref | |
| | Female | 1.03 (0.87–1.23, p=0.703) | | 1.10 (0.94–1.28, p=0.226) | | 0.97 (0.79–1.20, p=0.796) | |
| Household size(n) | 1 | ref | ref | ref | ref | ref | ref |
| | 2 | 0.41 (0.23–0.71, p=0.002) | 0.54 (0.29–0.99, p=0.048) | 0.37 (0.21–0.65, p<0.001) | 0.46 (0.25–0.85, p=0.013) | 0.62 (0.34–1.15, p=0.133) | 0.87 (0.45–1.69, p=0.686) |
| | 3 | 0.34 (0.20–0.57, p<0.001) | 0.50 (0.28–0.90, p=0.020) | 0.33 (0.19–0.57, p<0.001) | 0.44 (0.25–0.80, p=0.007) | 0.40 (0.22–0.71, p=0.002) | 0.59 (0.31–1.11, p=0.103) |
| | ≥4 | 0.38 (0.23–0.63, p<0.001) | 0.55 (0.31–0.97, p=0.041) | 0.35 (0.20–0.59, p<0.001) | 0.46 (0.26–0.81, p=0.008) | 0.45 (0.25–0.78, p=0.005) | 0.67 (0.36–1.24, p=0.198) |
| Domicile | Village/Rural | ref | ref | ref | ref | ref | |
| | City | 0.82 (0.69–0.98, p=0.028) | 0.87 (0.70–1.07, p=0.191) | 0.70 (0.60–0.81, p<0.001) | 0.83 (0.69–1.00, p=0.046) | 0.94 (0.76–1.15, p=0.535) | |
| **Academic Factors** | | | | | | | |
| Grade | Junior | ref | | ref | ref | ref | ref |
| | Senior | 1.57 (1.28–1.92, p<0.001) | 0.65 (0.37–1.13, p=0.129) | 1.41 (1.19–1.66, p<0.001) | 0.10 (0.06–0.17, p<0.001) | 1.35 (1.06–1.71, p=0.014) | 0.92 (0.48–1.76, p=0.801) |
| Learning styles | commuter learning | ref | ref | ref | ref | ref | ref |
| | residential learning | 0.78 (0.64–0.95, p=0.012) | 0.78 (0.62–0.99, p=0.039) | 1.27 (1.07–1.52, p=0.007) | 1.23 (1.00–1.50, p=0.048) | 0.84 (0.66–1.05, p=0.122) | 0.88 (0.69–1.12, p=0.285) |
| grade ranking | Top 10% | ref | ref | ref | | ref | |
| | 11%−30% | 1.10 (0.81–1.48, p=0.548) | 1.13 (0.82–1.56, p=0.463) | 1.02 (0.80–1.31, p=0.856) | | 0.84 (0.59–1.18, p=0.307) | |
| | 31%−60% | 1.31 (0.98–1.76, p=0.072) | 1.20 (0.87–1.65, p=0.260) | 0.94 (0.74–1.20, p=0.621) | | 0.98 (0.70–1.36, p=0.896) | |
| | < 60% | 1.80 (1.33–2.44, p<0.001) | 1.62 (1.17–2.25, p=0.004) | 0.95 (0.73–1.23, p=0.691) | | 1.22 (0.87–1.71, p=0.256) | |
| **Psychosocial Factors** | | | | | | | |
| Family Support | seldom | ref | ref | ref | | ref | |
| | average | 0.46 (0.33–0.65, p<0.001) | 0.66 (0.45–0.97, p=0.034) | 0.67 (0.47–0.94, p=0.020) | 0.85 (0.58–1.24, p=0.400) | 0.42 (0.29–0.60, p<0.001) | 0.60 (0.40–0.91, p=.016) |
| | More | 0.22 (0.16–0.31, p<0.001) | 0.45 (0.30–0.68, p<0.001) | 0.38 (0.27–0.53, p<0.001) | 0.69 (0.47–1.01, p=0.058) | 0.26 (0.18–0.37, p<0.001) | 0.53 (0.35–0.82, p=0.004) |
| Peer Support | seldom | ref | ref | ref | | ref | |
| | average | 0.41 (0.27–0.60, p<0.001) | 0.65 (0.41–1.02, p=0.062) | 0.67 (0.45–1.00, p=0.051) | 0.90 (0.58–1.41, p=0.648) | 0.41 (0.27–0.62, p<0.001) | 0.66 (0.41–1.06, p=0.086) |
| | More | 0.27 (0.18–0.39, p<0.001) | 0.64 (0.41–1.00, p=0.050) | 0.41 (0.28–0.59, p<0.001) | 0.73 (0.47–1.12, p=0.151) | 0.28 (0.19–0.41, p<0.001) | 0.57 (0.36–0.91, p=0.019) |

*(Continued)*

**Table 3.** (Continued)

| Variables | Levels | Depression | | Anxiety | | Stress | |
|---|---|---|---|---|---|---|---|
| | | Model 1[a] | Model 2[b] | Model 1 [a] | Model 2[b] | Model 1 [a] | Model 2 [b] |
| Help-seeking behavior | Passive | ref | ref | ref | | ref | |
| | Reactive | 0.35 (0.26–0.47, p<0.001) | 0.81 (0.56–1.16, p=0.252) | 0.52 (0.39–0.71, p<0.001) | 1.02 (0.71–1.48, p=0.899) | 0.35 (0.25–0.48, p<0.001) | 0.74 (0.50–1.09, p=0.133) |
| | Proactive | 0.17 (0.12–0.24, p<0.001) | 0.48 (0.32–0.73, p<0.001) | 0.31 (0.22–0.42, p<0.001) | 0.70 (0.47–1.04, p=.077) | 0.23 (0.16–0.33, p<0.001) | 0.58 (0.37–0.92, p=0.020) |
| | Low need | 0.10 (0.07–0.15, p<0.001) | 0.33 (0.21–0.52, p<0.001) | 0.21 (0.15–0.29, p<0.001) | 0.52 (0.34–0.79, p=0.002) | 0.14 (0.09–0.21, p<0.001) | 0.43 (0.26–0.71, p=0.001) |
| Preferred confidant | No confidant | ref | ref | ref | ref | ref | ref |
| | Family | 0.12 (0.08–0.16, p<0.001) | 0.22 (0.15–0.33, p<0.001) | 0.19 (0.14–0.26, p<0.001) | 0.27 (0.19–0.40, p<0.001) | 0.13 (0.09–0.20, p<0.001) | 0.23 (0.15–0.36, p<0.001) |
| | Peers | 0.24 (0.18–0.32, p<0.001) | 0.37 (0.26–0.53, p<0.001) | 0.33 (0.25–0.45, p<0.001) | 0.42 (0.29–0.60, p<0.001) | 0.31 (0.23–0.43, p<0.001) | 0.48 (0.33–0.70, p<0.001) |
| | Specific close person | 0.36 (0.25–0.51, p<0.001) | 0.56 (0.38–0.84, p=0.005) | 0.41 (0.28–0.58, p<0.001) | 0.55 (0.37–0.83, p=0.004) | 0.46 (0.32–0.67, p<0.001) | 0.70 (0.46–1.06, p=0.094) |
| | Low need | 0.13 (0.09–0.18, p<0.001) | 0.24 (0.16–0.35, p<0.001) | 0.21 (0.16–0.29, p<0.001) | 0.32 (0.22–0.47, p<0.001) | 0.13 (0.09–0.20, p<0.001) | 0.23 (0.15–0.35, p<0.001) |

[a] Model 1: univariate binary logistic regression analysis.

[b] Model 2: Based on the results of Model 1, to address multicollinearity, variables with a p-value greater than 0.1 or a variance inflation factor (VIF) exceeding 10 were excluded from the analysis.

3.2 tests per week result in 70.1% of students reporting significant somatisation symptoms [27]. Therefore, greater attention should be paid to the mental health of high school students, appropriate psychological counselling activities should be conducted, and a home-school collaborative psychological support network should be established to block the transmission of stress and alleviate their depression, anxiety, and stress symptoms.

In the study, we found that students from single-parent families were more likely to experience depression, anxiety, and stress symptoms compared to those from two-parent families. These students often bear heavier psychological pressure when facing changes in family structure than their peers. Huang Xinxin et al [28]. showed that parents and grandparents have a certain positive impact in family structure, and family structure can affect the occurrence of adolescent depressive symptoms. The environment of single-parent family may lead students to lack stable emotional support and sense of security, thereby triggering depression, anxiety and other emotional problems [29]. At the same time, the prejudice and misunderstanding of society towards single-parent families may also leave these students feeling isolated and helpless, further exacerbating their psychological burden [30]. Furthermore, middle school students are at a critical period of physical and mental development, and their ability to recognize themselves and regulate emotions is not yet fully mature, which makes them more susceptible to external factors and prone to mental health problems.

When middle school students encounter psychological disturbances, they often lack effective channels for seeking assistance. On the one hand, in a dictatorial family model where the family environment lacks openness and inclusiveness, poor communication between parents and children makes it challenging for children to obtain understanding and support within the family. On the other hand, peer relationships may also pose an impediment to students' seeking of help. As middle school students are in a crucial period of establishing interpersonal relationships, phenomena such as competition and exclusion among peers occur frequently, presenting numerous difficulties for students when seeking peer support. Additionally, students' own psychological factors are also significant elements influencing their help-seeking behaviors. Some students might have overly strong self-esteem and be reluctant to reveal their vulnerability to

others; others might lack a correct perception of mental health issues, considering seeking help as a sign of weakness, and thus choose to endure the pain alone. Therefore, in order to effectively enhance the mental health conditions of middle school students, we need to commence from multiple levels such as family, school, and society, and construct a more comprehensive support system to assist students in promptly identifying and addressing mental health problems.

There are some limitations to this study, firstly, the cross-sectional design used in this study can only provide a snapshot of a specific time period, and the relationship between environmental influences and psychological status needs to be further validated; Second, the number of middle school students surveyed was not as large as that of high school students because most middle school students did not wear cell phones themselves, and the number of middle school students overall was also called less than that of high school students; Third, no family history of depression or any mood disorders were obtained during the survey of secondary school students, which may have biased this study, and despite these biases, this study provides useful data and recommendations for schools to guide secondary school students on their mental health status in the future.

In summary, the results of this study show that the psychological status of middle school students is closely related to grade level, academic performance, family psychological support, and psychological support from peers.

## Acknowledgments

We would like to thank the teachers and students of Ya'an Middle School for their help in the data collection process.

## Author contributions

**Data curation:** Tingting Dong, Yumei Wang, Yuanjie Lin.

**Formal analysis:** Tingting Dong, Yumei Wang.

**Funding acquisition:** Tingting Dong.

**Investigation:** Tingting Dong, Yumei Wang, Yuanjie Lin.

**Methodology:** Tingting Dong, Yumei Wang, Yuanjie Lin.

**Project administration:** Tingting Dong.

**Resources:** Tingting Dong, Yumei Wang.

**Supervision:** Tingting Dong, Yumei Wang.

**Validation:** Tingting Dong.

**Visualization:** Tingting Dong.

**Writing – original draft:** Tingting Dong, Yumei Wang, Yuanjie Lin.

**Writing – review & editing:** Tingting Dong, Yumei Wang.

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
