## [Decision Letter · Decision Letter 0]

16 May 2025

Prevalence and Determinants of Depression, Anxiety, and Stress Among Secondary School Students

PLOS ONE

Thank you for submitting your manuscript to PLOS ONE. After careful consideration, we feel that it has merit but does not fully meet PLOS ONE’s publication criteria as it currently stands. Therefore, we invite you to submit a revised version of the manuscript that addresses the points raised during the review process.

Please submit your revised manuscript by Jun 28 2025 11:59PM  If you will need more time than this to complete your revisions, please reply to this message or contact the journal office at plosone@plos.org . A rebuttal letter that responds to each point raised by the academic editor and reviewer(s). You should upload this letter as a separate file labeled 'Response to Reviewers'.A marked-up copy of your manuscript that highlights changes made to the original version. You should upload this as a separate file labeled 'Revised Manuscript with Track Changes'.An unmarked version of your revised paper without tracked changes. You should upload this as a separate file labeled 'Manuscript'.If applicable, we recommend that you deposit your laboratory protocols in protocols.io to enhance the reproducibility of your results. Protocols.io assigns your protocol its own identifier (DOI) so that it can be cited independently in the future. For instructions see: https://journals.plos.org/plosone/s/submission-guidelines#loc-laboratory-protocols . Additionally, PLOS ONE offers an option for publishing peer-reviewed Lab Protocol articles, which describe protocols hosted on protocols.io. Read more information on sharing protocols at https://plos.org/protocols?utm_medium=editorial-email&utm_source=authorletters&utm_campaign=protocols .

We look forward to receiving your revised manuscript

Kind regards,

Maria José Nogueira, Ph.D.

Academic Editor

PLOS ONE

“This study was supported by a research project grant from the Chengdu Medical Association (Project No. 2024350).”

4. We note that your Data Availability Statement is currently as follows: All relevant data are within the manuscript and in Supporting Information files.

Additional Editor Comments (if provided):

Dear Author,

The study is very relevant, but it needs to be improved to provide greater robustness and clarity in the results.

Some suggestions for improvement

1-Tables with characterization data should appear before tables with information on other variables (depression, etc.)

2- tables have a lot of information, difficult to analyze. Improve the clarity/synthesis of information in tables, remove the synthesis of what is absolutely fundamental/use the body text or footnotes to give additional information, e.g. acronyms used for instruments; better identify the dimensions of the instruments...

Reviewers' comments:

Reviewer's Responses to Questions

**Comments to the Author**

1. Is the manuscript technically sound, and do the data support the conclusions?

Reviewer #1: Partly

2. Has the statistical analysis been performed appropriately and rigorously?

Reviewer #1: Yes

3. Have the authors made all data underlying the findings in their manuscript fully available?

Reviewer #1: Yes

4. Is the manuscript presented in an intelligible fashion and written in standard English?

Reviewer #1: Yes

Reviewer #1: excellent study but needs to be improved for greater impact.

in the abstract, presents a conclusion that is not found at the end of the article. You should revise this conclusion in line with the aim of the study, or just mainted the discussion. The key words should differentiate the type of student: high school? allows the article to be found in another way.

In the introduction:

line 52 - the author does not match the bibliographical references.

It is important to have a brief definition of adolescence and the ages to which this study refers. See the WHO definition in terms of age. They should also mention the risk factors that are already evident in adolescence. In order for the article to be international, it should mention the Chinese education system, since the results cover middle school and high school. This will be a more robust introduction and will allow us to understand the results presented.

In the methodology, they should be more specific, between what ages the study is carried out, the context (in the results they present middle school, high school and boarding school) and the sample in each context, so that we can understand the results.

On the DASS-21 scale, they should mention whether it has been validated for the Chinese population. There are always cultural adaptations.

in 3.2 they address the factors that influence middle school, there is no mention in the methodology of the level of education, the context and they should present the most relevant data in narrative form.

In 3.3 it is important to analyse the data in a narrative way, especially the main data.

in the discussion address something extremely important and which should be mentioned in the introduction when addressing the school system. The fact that high school students have an exam that leads to high levels of stress and anxiety should be explored more, as it is an important fact that can justify the figures.

**Do you want your identity to be public for this peer review?** For information about this choice, including consent withdrawal, please see our Privacy Policy

Reviewer #1: **Yes: ** Patricia Alves

---

## [Author Response · Author response to Decision Letter 1]

24 Jun 2025

Dear Editor and Reviewers,

We are very grateful to you and the reviewers for your important comments and thoughtful suggestions on our manuscript titled ‘[Prevalence and Determinants of Depression, Anxiety, and Stress Among Secondary School Students]’. Based on these comments and suggestions, we have carefully revised the original manuscript. All changes made to the text in the revised manuscript are marked with revision marks for easy identification. We would like to express our sincere gratitude once again for your comments and constructive suggestions, which are very valuable in improving the quality of our manuscript. Below are our responses to each of the reviewers' comments.

Response to Journal requirements

1.Please ensure that your manuscript meets PLOS ONE's style requirements, including those for file naming. The PLOS ONE style templates can be found at https://journals.plos.org/plosone/s/file?id=wjVg/PLOSOne_formatting_sample_main_body.pdf and

Response Thank you for your professional advice. We have revised the manuscript according to the format requirements of PLOS ONE.

Response Thank you very much for your suggested revisions. I have carefully reviewed the manuscript and removed all grant-related text to ensure that the manuscript no longer contains any funding information.

“This study was supported by a research project grant from the Chengdu Medical Association (Project No. 2024350).”

Please state what role the funders took in the study. If the funders had no role, please state: "The funders had no role in study design, data collection and analysis, decision to publish, or preparation of the manuscript."If this statement is not correct you must amend it as needed.

Response Thank you for your professional advice. Regarding the role of funders, we hereby state the following:

‘This study was funded by the Chengdu Medical Association Research Project (Project Number: 2024350). The funders had no role in study design, data collection and analysis, decision to publish, or preparation of the manuscript. The research team independently completed all scientific research work and is solely responsible for the content of the paper.’ This statement has been added to Section 4 ‘Funding Declaration’ of the submission cover letter.

The revised cover letter has been resubmitted as an attachment. We will strictly adhere to academic integrity standards to ensure the independence of the research.

4. We note that your Data Availability Statement is currently as follows: All relevant data are within the manuscript and in Supporting Information files.

Response Thank you for your professional advice. We have improved our data sharing mechanism in accordance with the PLOS Minimum Dataset standards and uploaded the data from this study as an attachment. The data includes the original DASS-21 scores for all subjects analysed and complete basic information.

5. Tables with characterization data should appear before tables with information on other variables (depression, etc.)

Response Thank you for pointing out the problem with the ordering of the tables. We have adjusted the ordering of the tables as requested, and adjusted the table of characterisation data (formerly Table 2) to Table 1, with the subsequent DASS-21 scores and distributional features numbered accordingly, as detailed in the manuscript line203-285.

6. tables have a lot of information, difficult to analyze. Improve the clarity/synthesis of information in tables, remove the synthesis of what is absolutely fundamental/use the body text or footnotes to give additional information, e.g. acronyms used for instruments; better identify the dimensions of the instruments...

Response Thank you for your comments on the presentation of the table. We have systematically optimised the table by restructuring the logical framework through hierarchical grouping of variables (demographics → academic factors → psychosocial factors), simplifying the nomenclature of key variables (e.g., simplifying ‘Taking the initiative to seek help...’ to ‘Help-seeking behaviour’), and adopting a standardised presentation of effect sizes. (e.g. ‘Taking the initiative to seek help or talk about one's problems’ was simplified to ‘Help-seeking behaviour’), which significantly improved the readability and analytical efficiency of the table while fully retaining the original data, see lines 256-262 for details of the modifications.

Response to Reviewer #1

1. in the abstract, presents a conclusion that is not found at the end of the article. You should revise this conclusion in line with the aim of the study, or just mainted the discussion. The key words should differentiate the type of student: high school? allows the article to be found in another way.

Response Thank you for your valuable comments. We have revised the abstract conclusion section and keywords to ensure consistency with the full discussion. The revised abstract conclusion is as follows:

“This study demonstrates that high school student populations, students with academic rankings <60%, and students who lack family/peer psychosocial support are at higher risk for mental health. These findings provide an empirical basis for the development of tiered intervention strategies for specific risk factors.”

2. line 52 - the author does not match the bibliographical references.

Response Thank you for pointing out the inconsistency between the authors cited in the text and the list of references. We have checked and corrected the citations (see line 67 for details), and the revised list of references has been highlighted in yellow in the revised version (see line 488-490 for details).

3. It is important to have a brief definition of adolescence and the ages to which this study refers. See the WHO definition in terms of age. They should also mention the risk factors that are already evident in adolescence. In order for the article to be international, it should mention the Chinese education system, since the results cover middle school and high school. This will be a more robust introduction and will allow us to understand the results presented.

Response Thank you for your important revision suggestions. We have revised the relevant content of the paper, firstly, adding the WHO definition of adolescence (10-19 years old) to make it clear that the target population of this study (12-18 years old) covers 86% of the adolescent stage; secondly, we have supplemented the key features of the Chinese education system, including the division of the academic system into junior high school (grades 7-9, 12-15 years old) and senior high school (grades 10-12, 16-18 years old), and the two key examination nodes, the secondary school examination and the two key examination nodes of the college entrance examination; and refined the discussion of typical risk factors in adolescence, including academic pressure, family relationships, and peer competition. All revisions were supported by the literature and clearly labelled in the revised draft. These revisions complete the context of the study and make it more accessible to an international audience. The revisions are detailed in lines 53-65.

4. In the methodology, they should be more specific, between what ages the study is carried out, the context (in the results they present middle school, high school and boarding school) and the sample in each context, so that we can understand the results.

Response Thank you for your important suggestions regarding the methodological rigour of the study, and we have systematically improved the description of the study population. The addition of a new section in the Methods section clarifies the age range of the study population (12-18 years old) in correspondence with the educational stage (grades 7-9 in middle school/grades 10-12 in high school), and these modifications make the correlation between the sample's characteristics and the educational context more transparent and easier for people to understand. See lines 103-110 of the manuscript for details.

5. On the DASS-21 scale, they should mention whether it has been validated for the Chinese population. There are always cultural adaptations.

Response Thank you for your important suggestions. We have supplemented and refined the use of the DASS-21 scale to clarify the validation study of the Chinese version of the scale in a Chinese population[1]. The trial (n=1779) showed good reliability of the scale (Cronbach's α for depression/anxiety/stress subscales of 0.77/0.79/0.76, respectively, and validated factor analytic fitness CFI=0.88, RMSEA=0.065), and was compared with the Beck Depression Inventory (BDI) (r=0.598), State Anxiety (SAI) (r=0.531) and Trait Anxiety (TAI) (r=0.552) presenting clinically significant validity scale associations. These revisions are detailed in section 2.4.2 of the Methods section (lines 136-151).

Reference

1. Gong X, Xie X, Xu R, Luo Y. Report on the Depression-Anxiety-Stress Scale in Simplified Chinese (DASS-21) among Chinese University Students. Chinese Journal of Clinical Psychology. 2010. doi: 10.16128/j.cnki.1005-3611.2010.04.020.

6. in 3.2 they address the factors that influence middle school, there is no mention in the methodology of the level of education, the context and they should present the most relevant data in narrative form.

Response Thank you for your suggestions. We have revised the research methods and results sections of the paper.

1. In the methods section, we have added a clear definition of the age range of the research subjects. According to the Chinese educational stage classification standards, junior high school corresponds to grades 7 to 9 (ages 12 to 15), and senior high school corresponds to grades 10 to 12 (ages 16 to 18). These revisions make the relationship between sample characteristics and educational background more transparent and easier to understand. See lines 103-110 of the manuscript.

2. In the research results section (Sections 3.1–3.2), we have revised the presentation of the research findings, using a consistent comparison benchmark (replacing inconsistent terminology with ‘vs’) and presenting the percentage of participants at risk of depression, anxiety, and stress. This approach highlights the most relevant data and presents the results in a more concise and clear manner. Specific revisions are detailed in lines 203–285.

7. In 3.3 it is important to analyse the data in a narrative way, especially the main data.

Response Thank you for your professional suggestions. We have made the following revisions to the results section of Section 3.3:

1. The data in Table 3 has been reclassified and reorganised to clearly present key results under three major categories: Demographics, Academic Factors, and Psychosocial Factors.

2. In the textual description of the results, we have adjusted the order of presentation, prioritising Psychosocial Factors, followed by Academic Factors, and finally Demographics, to highlight the core findings more effectively.

3. When presenting key results, we have abandoned the single reporting of raw p-values and instead emphasised effect sizes (odds ratios, OR) to more directly reflect the clinical significance and effect strength of the results.

These revisions aim to significantly enhance the clarity and coherence of the results. Specific modifications are detailed in lines 286 to 324 of the manuscript.

8. in the discussion address something extremely important and which should be mentioned in the introduction when addressing the school system. The fact that high school students have an exam that leads to high levels of stress and anxiety should be explored more, as it is an important fact that can justify the figures.

Response Thank you for your professional advice. Based on your suggestions, we have made revisions in the following three areas:

1. In the discussion section, we have provided further elaboration on the background of China's college entrance examination system as the core mechanism for talent selection within the higher education system. We have emphasized how this system systematically constructs sources of pressure for high school students through dimensions such as competitive mechanisms, assessment standards, and time management. As mentioned in the draft, elements such as the ‘National College Entrance Examination’ and the ‘12-hour schedule’ all reflect the role of the school system in shaping student stress.

2. We have cited data such as ‘over 10 million high school students compete for 600,000 to 800,000 spots in the college entrance examination each year’ and ‘gross enrolment rate in higher education,’ combined with micro-level research findings such as ‘a 32.7% increase in anxiety risk’ and ‘70.1% of students exhibiting somatisation symptoms’ to provide multi-dimensional data support. This data illustrates the impact of examination pressure on mental health. Additionally, physiological mechanisms such as the activation of the hypothalamic-pituitary-adrenal axis are introduced to enhance the scientific rigor of the argument.

3. Further elaboration on the interaction between family and school, presenting data such as ‘admission rates in major college entrance examination provinces like Jilin’ and ‘proportion of parents with academic expectations,’ clearly outlining the complete argumentative chain of ‘institutional pressure - family expectations - individual psychological responses,’ and explaining the connection between exam pressure and high prevalence rates of depression and anxiety symptoms. For detailed content, please refer to lines 333-391 of the manuscript.

We sincerely appreciate the time and effort that the reviewers have invested in evaluating our manuscript. We are eager to receive any additional feedback or suggestions that may further enhance our work.

Tingting Dong

---

## [Decision Letter · Decision Letter 1]

8 Jul 2025

Prevalence and Determinants of Depression, Anxiety, and Stress Among Secondary School Students

PONE-D-25-01533R1

Dear Dr. tingting dong

We’re pleased to inform you that your manuscript has been judged scientifically suitable for publication and will be formally accepted for publication once it meets all outstanding technical requirements.

Kind regards,

Maria José Nogueira, Ph.D.

Academic Editor

PLOS ONE

Additional Editor Comments (optional):

The authors considered the proposed suggestions for improvement, which added greater robustness to the work, which can be accepted for publication.

Reviewers' comments:

Reviewer's Responses to Questions

**Comments to the Author**

Reviewer #1: All comments have been addressed

2. Is the manuscript technically sound, and do the data support the conclusions?

Reviewer #1: Yes

3. Has the statistical analysis been performed appropriately and rigorously?

Reviewer #1: Yes

4. Have the authors made all data underlying the findings in their manuscript fully available?

Reviewer #1: Yes

5. Is the manuscript presented in an intelligible fashion and written in standard English?

Reviewer #1: Yes

Reviewer #1: After the revision, the article is very interesting and worth reading. Results that should be valued

**Do you want your identity to be public for this peer review?** For information about this choice, including consent withdrawal, please see our Privacy Policy

Reviewer #1: **Yes: ** Patricia Alves

---

## [Editor Report · Acceptance letter]

PONE-D-25-01533R1

PLOS ONE

Dear Dr. dong,

I'm pleased to inform you that your manuscript has been deemed suitable for publication in PLOS ONE. Congratulations! Your manuscript is now being handed over to our production team.

Kind regards,

on behalf of

Professor Maria José Nogueira

Academic Editor

PLOS ONE